# A complex hydride lithium superionic conductor for high-energy-density all-solid-state lithium metal batteries

Sangryun Kim [1], Hiroyuki Oguchi[2], Naoki Toyama[1], Toyoto Sato [1], Shigeyuki Takagi[1], Toshiya Otomo[3], Dorai Arunkumar [4], Naoaki Kuwata[4], Junichi Kawamura[4] & Shin-ichi Orimo[1,2]

All-solid-state batteries incorporating lithium metal anode have the potential to address the energy density issues of conventional lithium-ion batteries that use flammable organic liquid electrolytes and low-capacity carbonaceous anodes. However, they suffer from high lithium ion transfer resistance, mainly due to the instability of the solid electrolytes against lithium metal, limiting their use in practical cells. Here, we report a complex hydride lithium superionic conductor, $0.7Li(CB_9H_{10})$–$0.3Li(CB_{11}H_{12})$, with excellent stability against lithium metal and a high conductivity of $6.7 \times 10^{-3}$ S cm$^{-1}$ at 25 °C. This complex hydride exhibits stable lithium plating/stripping reaction with negligible interfacial resistance (<1 Ω cm$^2$) at 0.2 mA cm$^{-2}$, enabling all-solid-state lithium-sulfur batteries with high energy density (>2500 Wh kg$^{-1}$) at a high current density of 5016 mA g$^{-1}$. The present study opens up an unexplored research area in the field of solid electrolyte materials, contributing to the development of high-energy-density batteries.

[1] Institute for Materials Research, Tohoku University, 2-1-1 Katahira, Aoba-ku, Sendai 980-8577, Japan. [2] WPI-Advanced Institute for Materials Research (WPI-AIMR), Tohoku University, 2-1-1 Katahira, Aoba-ku, Sendai 980-8577, Japan. [3] Institute of Materials Structure Science, High Energy Accelerator Research Organization, 1-1 Oho, Tsukuba 305-0801, Japan. [4] Institute of Multidisciplinary Research for Advanced Materials, Tohoku University, 2-1-1 Katahira, Aoba-ku, Sendai 980-8577, Japan. Correspondence and requests for materials should be addressed to S.K. (email: sangryun@imr.tohoku.ac.jp) or to S.-i.O. (email: orimo@imr.tohoku.ac.jp)

All-solid-state batteries are promising candidates for resolving the intrinsic drawbacks of current lithium-ion batteries, such as electrolyte leakage, flammability, and limited energy density[1–3]. Recent extensive research on all-solid-state batteries has led to considerable progress in solid electrolytes, which are generally categorized into sulfide solid electrolytes and oxide solid electrolytes. The most attractive feature of sulfide solid electrolytes, such as $Li_{10}GeP_2S_{12}$-type compounds[1,4,5], $Li_2S–P_2S_5$ glass ceramics[2,6], and argyrodites[7,8], is their high lithium ion conductivity of over $10^{-3}$ S cm$^{-1}$ at room temperature, which is comparable to those of liquid electrolytes. Oxide solid electrolytes such as perovskite-type[9,10] and garnet-type[11,12] materials have been also found to be promising due to their high processing flexibility and air stability.

Despite tremendous research advancements in solid electrolytes, the development of all-solid-state batteries for practical applications, such as electric vehicles and grid-scale energy storage systems, is still in its infancy, mainly in terms of energy density. This has stimulated research into combining suitable high-energy-density electrodes with solid electrolytes. In this regard, lithium metal is the ultimate anode material for all-solid-state batteries because it has the highest theoretical capacity (3860 mAh g$^{-1}$) and the lowest potential (−3.04 V vs. standard hydrogen electrode) among known anode materials. However, most existing solid electrolytes have chemical/electrochemical instability and/or poor physical contact against lithium metal, inevitably causing unwanted side reactions at the interface[13–15]. These side reactions result in an increase in interfacial resistance, greatly degrading battery performance during repeated cycling. Efforts have been made to overcome these shortcomings, including alloying the lithium metal anode[16,17] and introducing buffer layers[3,18,19]. However, lithium metal alloys have higher potential than that of pure lithium metal, lowering the cell voltage and thus energy density. In addition, buffer layers increase cell resistance due to their lower conductivities compared to those of solid electrolytes. It is thus desirable to find a solid electrolyte that is intrinsically stable and compatible with lithium metal to maximize the advantages of the lithium metal anode.

Complex hydrides, generally denoted as $M_x(M'_yH_z)$, where $M$ represents a metal cation and $M'_yH_z$ represents a complex anion, have received particular attention as a new class of solid electrolytes to address the problems associated with the lithium metal anode owing to their high deformability and outstanding chemical/electrochemical stability against the lithium metal anode, which results from their high reducing ability[20,21]. However, the major drawback of complex hydrides is their low ionic conductivity (~$10^{-5}$ S cm$^{-1}$ at room temperature), thus requiring high-temperature (~100 °C) operation for stable battery performance[20,22,23]. Therefore, the development of complex hydride solid electrolytes that exhibit high ionic conductivity at room temperature will be a revolutionary breakthrough for all-solid-state batteries employing a lithium metal anode.

In this work, we develop a complex hydride lithium superionic conductor from a solid solution of two complex hydrides, namely $Li(CB_9H_{10})$ and $Li(CB_{11}H_{12})$. The partial replacement of $(CB_9H_{10})^-$ with $(CB_{11}H_{12})^-$ stabilizes the disordered high-temperature (high-$T$) phase of $Li(CB_9H_{10})$ at lower temperatures, leading to a lithium superionic conductivity of $6.7 \times 10^{-3}$ S cm$^{-1}$ at 25 °C. This material shows unparalleled stability with lithium metal and thus has a variety of advantages as the solid electrolyte for all-solid-state lithium metal batteries even near room temperature, allowing high lithium ion transfer capability at the interface with the lithium metal anode and stable cycling of high-energy-density all-solid-state lithium–sulfur (Li–S) batteries.

## Results

**Material synthesis and characterization.** A complex hydride lithium superionic conductor was synthesized by stabilizing the disordered high-$T$ phase of a *closo*-type complex hydride containing *closo*-type (cage-like) complex anions (Supplementary Fig. 1 and Supplementary Note 1) at room temperature. In this study, $Li(CB_9H_{10})$ was chosen as the host starting material for the stabilization of the high-$T$ phase because of its low phase transition temperature (90 °C) and high lithium ion conductivity, approaching $10^{-1}$ S cm$^{-1}$ for the high-$T$ phase[24]. The stabilization of the high-$T$ phase of $Li(CB_9H_{10})$ was achieved by partially replacing $(CB_9H_{10})^-$ complex anions with $(CB_{11}H_{12})^-$ complex anions using a mechanical ball-milling technique (see the Methods section for details). $(CB_{11}H_{12})^-$ complex anions were used as they have similar geometry and size and the same valence compared to those of $(CB_9H_{10})^-$ complex anions. The phase transition temperatures and ionic conductivities of *closo*-type complex hydrides are summarized in Supplementary Table 1. Additional descriptions of the starting materials ($Li(CB_9H_{10})$ and $Li(CB_{11}H_{12})$), including the geometries of $(CB_9H_{10})^-$ and $(CB_{11}H_{12})^-$ complex anions (Supplementary Fig. 1), X-ray diffraction (XRD) patterns (Supplementary Fig. 2), field-emission scanning electron microscopy (FE-SEM) images (Supplementary Fig. 3), and differential thermal analysis (DTA) profiles (Supplementary Fig. 4), are provided in the Supplementary Information.

From the testing of various $(CB_{11}H_{12})^-$ complex anion content levels, a 0.3 molar fraction of $(CB_{11}H_{12})^-$ was chosen as the main composition (denoted as $0.7Li(CB_9H_{10})–0.3Li(CB_{11}H_{12})$) for stabilizing the high-$T$ phase of $Li(CB_9H_{10})$, as lower content (0.1 molar fraction) resulted in the incomplete stabilization of the high-$T$ phase and higher content (0.5 molar fraction) led to the formation of other impurity phases (Supplementary Fig. 5 and Supplementary Note 2). The XRD pattern of $0.7Li(CB_9H_{10})–0.3Li(CB_{11}H_{12})$ shows new diffraction peaks that are not assigned to the low-temperature (low-$T$) phases of the starting materials (Fig. 1a, Supplementary Figs. 6 and 7, and Supplementary Note 3). These peaks were indexed by the hexagonal unit cell, which is consistent with that (space group $P31c$ ($Z = 2$)[24]) of the high-$T$ phase of $Li(CB_9H_{10})$. The estimated unit cell volume per formula unit ($V/Z = 219$ Å$^3$) of $0.7Li(CB_9H_{10})–0.3Li(CB_{11}H_{12})$ is larger than that of $Li(CB_9H_{10})$ ($V/Z = 205$ Å$^3$) (Supplementary Table 2), revealing the formation of a solid-solution phase of $Li(CB_9H_{10})$ and $Li(CB_{11}H_{12})$, in which the centres of $(CB_9H_{10})^-$ and $(CB_{11}H_{12})^-$ occupy the same sites ($2b = (1/3, 2/3, z)$) in the crystal structure. The expanded lattice is consistent with the substitution of the larger $(CB_{11}H_{12})^-$ for the smaller $(CB_9H_{10})^-$. These results verify that the high-$T$ phase of $Li(CB_9H_{10})$ is stabilized at room temperature by the partial substitution of complex anions. The lowered phase transition temperature can be possibly attributed to the increased entropy change due to the effects of the formation of the solid-solution phase, by which disordered distributions of both lithium ions and complex anions are thermodynamically preferred.

The phase transition between the low-$T$ and high-$T$ phases was investigated by DTA (Fig. 1b and Supplementary Table 1). The DTA profile of $Li(CB_9H_{10})$ cycled between 25 and 200 °C exhibits endothermic and exothermic peaks at 90 and 80 °C upon heating–cooling cycling, which originate from the reversible transitions to and from the disordered high-$T$ phase[24]. Importantly, $0.7Li(CB_9H_{10})–0.3Li(CB_{11}H_{12})$ displays no such endothermic and exothermic peaks, reconfirming the stabilization of the high-$T$ phase at room temperature. Additionally, the high thermal stability of $0.7Li(CB_9H_{10})–0.3Li(CB_{11}H_{12})$ was verified by XRD measurements after the heat-treatment at 473 K for 12 h (Supplementary Fig. 8).

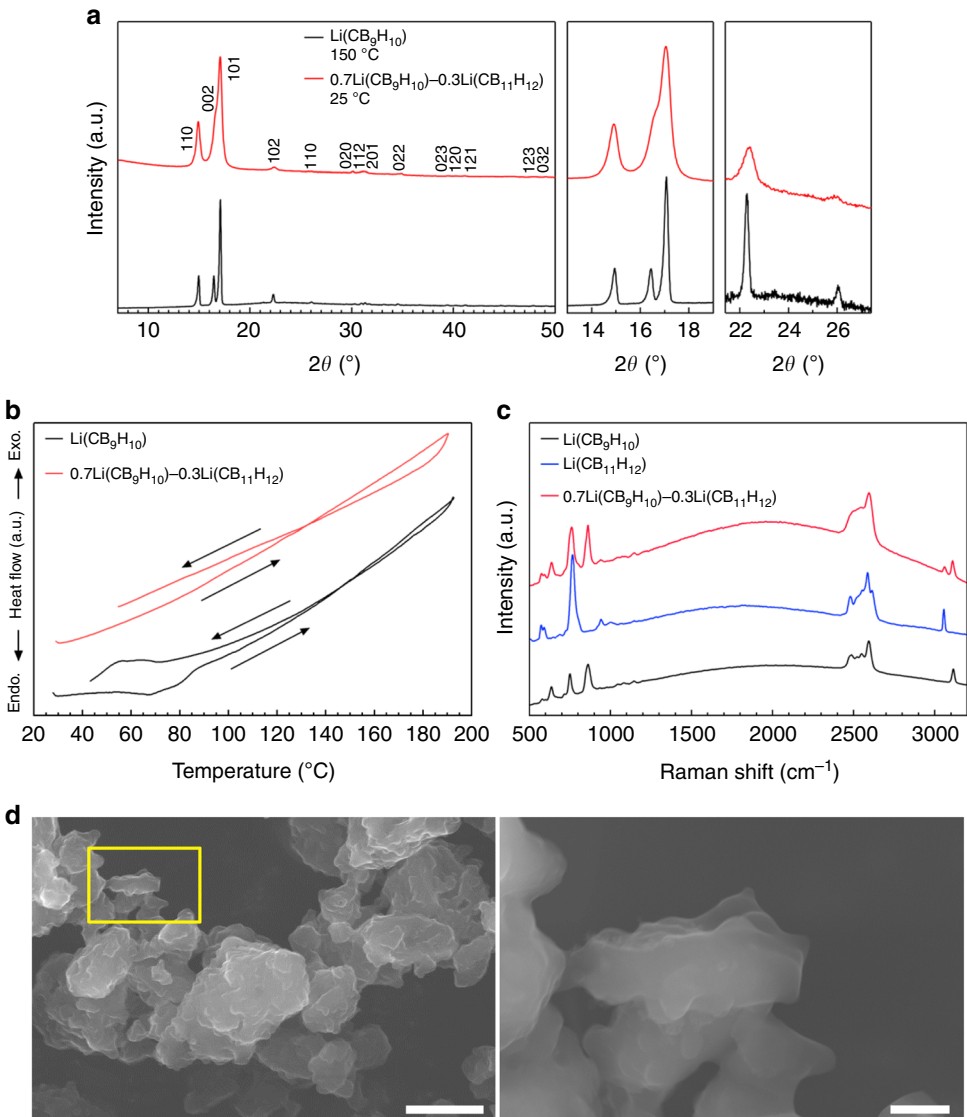

**Fig. 1** Stabilization of high-$T$ phase at room temperature. **a** XRD patterns of $Li(CB_9H_{10})$ at 150 °C and $0.7Li(CB_9H_{10})$–$0.3Li(CB_{11}H_{12})$ at room temperature. **b** DTA curves of $Li(CB_9H_{10})$ and $0.7Li(CB_9H_{10})$–$0.3Li(CB_{11}H_{12})$. **c** Raman spectra of $Li(CB_9H_{10})$, $Li(CB_{11}H_{12})$, and $0.7Li(CB_9H_{10})$–$0.3Li(CB_{11}H_{12})$. **d** FE-SEM images of $0.7Li(CB_9H_{10})$–$0.3Li(CB_{11}H_{12})$. The magnified image (right) is of the yellow marked area (left). Scale bars, 20 μm for the left image in **d** and 3 μm for the right image in **d**

The vibrational modes of complex anions in $0.7Li(CB_9H_{10})$–$0.3Li(CB_{11}H_{12})$ were examined by Raman spectroscopy measurements (Fig. 1c). In the low-Raman-shift region below 1200 cm$^{-1}$, the Raman profile of $0.7Li(CB_9H_{10})$–$0.3Li(CB_{11}H_{12})$ exhibits various deformation vibration modes[25] of both $(CB_9H_{10})^-$ and $(CB_{11}H_{12})^-$. In addition, Raman peaks at 3110 and 3050 cm$^{-1}$, which are ascribed to C–H stretching modes[25] in $(CB_9H_{10})^-$ and $(CB_{11}H_{12})^-$, respectively, were observed for $0.7Li(CB_9H_{10})$–$0.3Li(CB_{11}H_{12})$. These results indicate that $(CB_9H_{10})^-$ and $(CB_{11}H_{12})^-$ remain intact and coexist in the solid-solution phase.

FE-SEM images of $0.7Li(CB_9H_{10})$–$0.3Li(CB_{11}H_{12})$ depict that the prepared sample forms secondary particles with sizes of 10–20 μm consisting of primary particles of imperfect circular morphology with sizes of 1–3 μm (Fig. 1d and Supplementary Fig. 9). The primary particles were interconnected with very smooth edges. These morphologies reflect the softness and deformability of the prepared sample, which can allow close contact with electrode materials during cell preparation.

**Lithium ion conductivity**. The ionic conductivity of $0.7Li(CB_9H_{10})$–$0.3Li(CB_{11}H_{12})$ was assessed using electrochemical impedance spectroscopy (EIS) measurements with Au electrodes. For the measurements, the pelletized samples were prepared by cold-pressing at 153.6 MPa. To provide a reference for a quantitative comparison, the results for $Li(CB_9H_{10})$ are also shown[24]. The impedance profile of $Li(CB_9H_{10})$ at 25 °C (=298 K) exhibits one semicircle in the high-frequency region and one spike in the low-frequency region (Fig. 2a), which correspond to contributions from the bulk/grain boundary and the electrode, respectively. In contrast, $0.7Li(CB_9H_{10})$–$0.3Li(CB_{11}H_{12})$ displays only one spike, which is ascribed to the electrode contribution. The impedance of $0.7Li(CB_9H_{10})$–$0.3Li(CB_{11}H_{12})$, which includes bulk and grain boundary resistances, was determined from the intercept of the electrode spike on the $Z'$-axis. Of note, the impedance measured for $0.7Li(CB_9H_{10})$–$0.3Li(CB_{11}H_{12})$ is orders of magnitude lower compared to that of $Li(CB_9H_{10})$. The lithium ion conductivity ($\sigma$) at 25 °C of $0.7Li(CB_9H_{10})$–$0.3Li(CB_{11}H_{12})$ is $6.7 \times 10^{-3}$ S cm$^{-1}$, which is three

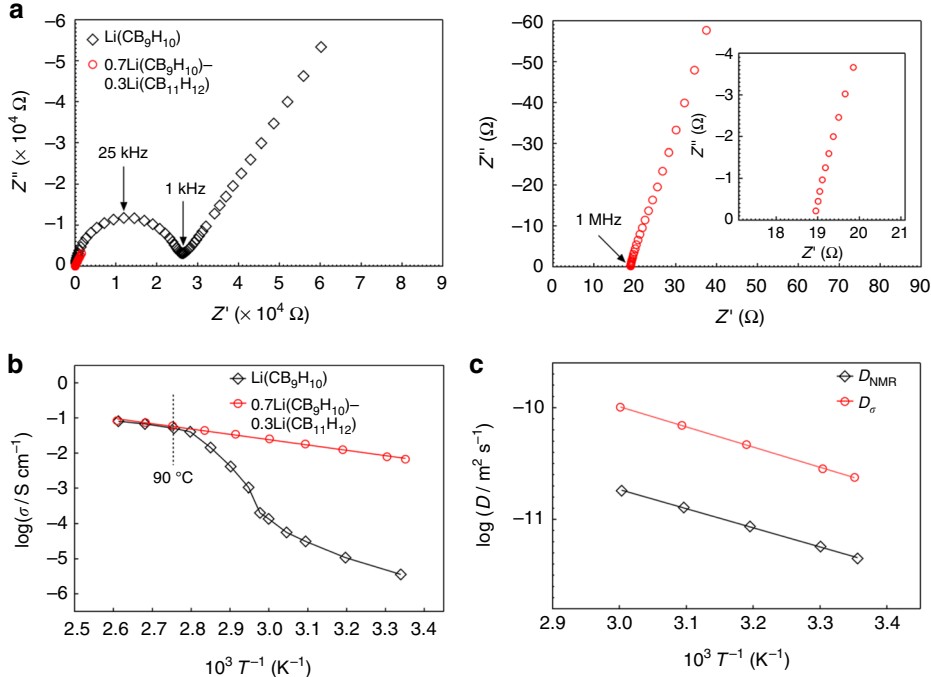

**Fig. 2** Lithium ion conductivity of $0.7Li(CB_9H_{10})$–$0.3Li(CB_{11}H_{12})$. **a** Nyquist plots of $Li(CB_9H_{10})$ and $0.7Li(CB_9H_{10})$–$0.3Li(CB_{11}H_{12})$ at 25 °C (left). Magnified Nyquist plots of $0.7Li(CB_9H_{10})$–$0.3Li(CB_{11}H_{12})$ in the high-frequency region (right). **b** Arrhenius plots of the lithium ion conductivities of $Li(CB_9H_{10})$ and $0.7Li(CB_9H_{10})$–$0.3Li(CB_{11}H_{12})$. **c** Arrhenius plots of the diffusion coefficients calculated from the impedance and NMR measurements

orders of magnitude higher than that ($\sigma = 3.6 \times 10^{-6}\,\mathrm{S\,cm^{-1}}$) of $Li(CB_9H_{10})$.

An Arrhenius plot of ionic conductivities shows that as the temperature increases from 25 to 90 °C, $Li(CB_9H_{10})$ displays a drastic jump in ionic conductivity, which originates from a transition to the high-$T$ phase (Fig. 2b, Supplementary Fig. 10a, and Supplementary Note 4). In contrast, for $0.7Li(CB_9H_{10})$–$0.3Li(CB_{11}H_{12})$, the Arrhenius plot shows no phase-transition-derived changes in conductivity and a linear increase in the logarithmic values (Fig. 2b and Supplementary Fig. 10b). Importantly, at above 90 °C, the conductivities of $0.7Li(CB_9H_{10})$–$0.3Li(CB_{11}H_{12})$ match well with those of $Li(CB_9H_{10})$: the conductivities at 110 °C and activation energies ($E_a$) of $Li(CB_9H_{10})$ and $0.7Li(CB_9H_{10})$–$0.3Li(CB_{11}H_{12})$ are $8.1 \times 10^{-2}\,\mathrm{S\,cm^{-1}}$ and 28.9 kJ mol⁻¹, and $8.5 \times 10^{-2}\,\mathrm{S\,cm^{-1}}$ and 28.4 kJ mol⁻¹, respectively (Supplementary Table 1 and Supplementary Note 5). It has been reported that the fast ionic conduction in the high-$T$ phase of complex hydrides results from their vacancy-rich disordered cation sublattices within networks of reorientationally disordered complex anions[24,26–29].

$^7Li$-pulsed field gradient nuclear magnetic resonance (PFG NMR) analyses indicate that the self-diffusion coefficient ($D_{NMR}$) of $0.7Li(CB_9H_{10})$–$0.3Li(CB_{11}H_{12})$ is lower than the conductivity diffusion coefficient ($D_\sigma$) calculated from the impedance measurements (Fig. 2c, Supplementary Fig. 11, and Supplementary Note 6). The Haven ratio, $H_R = D_{NMR}/D_\sigma$, was ~0.17 in the temperature range of 25–60 °C, clarifying that the high ionic conductivity of $0.7Li(CB_9H_{10})$–$0.3Li(CB_{11}H_{12})$ is the result of a strong correlation between highly disordered lithium ions, which induces concerted ionic diffusion[30–32].

To the best of our knowledge, the room-temperature conductivity ($6.7 \times 10^{-3}\,\mathrm{S\,cm^{-1}}$ at 25 °C) of $0.7Li(CB_9H_{10})$–$0.3Li(CB_{11}H_{12})$ is the highest value reported to date for complex hydride lithium ion conductors (Fig. 3). This conductivity is comparable to those of oxide-based[9,11] and sulfide-based[2,4,7] solid electrolytes. Considering that organic liquid electrolytes[33] have transport numbers of below 0.5,

$0.7Li(CB_9H_{10})$–$0.3Li(CB_{11}H_{12})$ has higher conductivity than organic liquid electrolytes.

**Stability against lithium metal anode.** $0.7Li(CB_9H_{10})$–$0.3Li(CB_{11}H_{12})$ shows markedly high stability against the lithium metal anode in terms of potential window, interfacial resistance, voltage polarization, and lithium plating/stripping cycling. Electrochemical stability was first evaluated by cyclic voltammetry (CV) with a Mo/$0.7Li(CB_9H_{10})$–$0.3Li(CB_{11}H_{12})$/Li cell at 25 °C (Fig. 4a and Supplementary Fig. 12). The cell displays sharp and reversible cathodic and anodic currents at around 0 V, which correspond to lithium deposition ($Li^+ + e^- \rightarrow Li$) and dissolution ($Li \rightarrow Li^+ + e^-$), respectively, revealing the high reducing ability of $0.7Li(CB_9H_{10})$–$0.3Li(CB_{11}H_{12})$. In addition, the lack of oxidation currents within the scanned voltage range (−0.1 to 5 V) demonstrates the wide potential window of $0.7Li(CB_9H_{10})$–$0.3Li(CB_{11}H_{12})$.

The interfacial resistance between $0.7Li(CB_9H_{10})$–$0.3Li(CB_{11}H_{12})$ and the lithium metal anode was investigated using the EIS measurement with a symmetric Li/$0.7Li(CB_9H_{10})$–$0.3Li(CB_{11}H_{12})$/Li cell at 25 °C (Fig. 4b). The impedance profile of the Li/$0.7Li(CB_9H_{10})$–$0.3Li(CB_{11}H_{12})$/Li cell displays one semicircle in the high-frequency region and one spike in the low-frequency region, which correspond to contributions from the $0.7Li(CB_9H_{10})$–$0.3Li(CB_{11}H_{12})$/Li interface and the electrode, respectively (Supplementary Fig. 13, Supplementary Table 3, and Supplementary Note 7). The $0.7Li(CB_9H_{10})$–$0.3Li(CB_{11}H_{12})$/Li interfacial resistance was calculated to be 0.78 Ω cm². This interfacial resistance is quite noticeable, as it is far lower than those of other solid electrolyte/Li interfaces reported to date[3,14,16,18,19,34–37]. The interfacial resistances between the solid electrolyte and the lithium metal anode are summarized in Supplementary Table 4. An FE-SEM image confirms close physical contact at the interface between $0.7Li(CB_9H_{10})$–$0.3Li(CB_{11}H_{12})$ and lithium metal (Fig. 4c). The remarkable interfacial

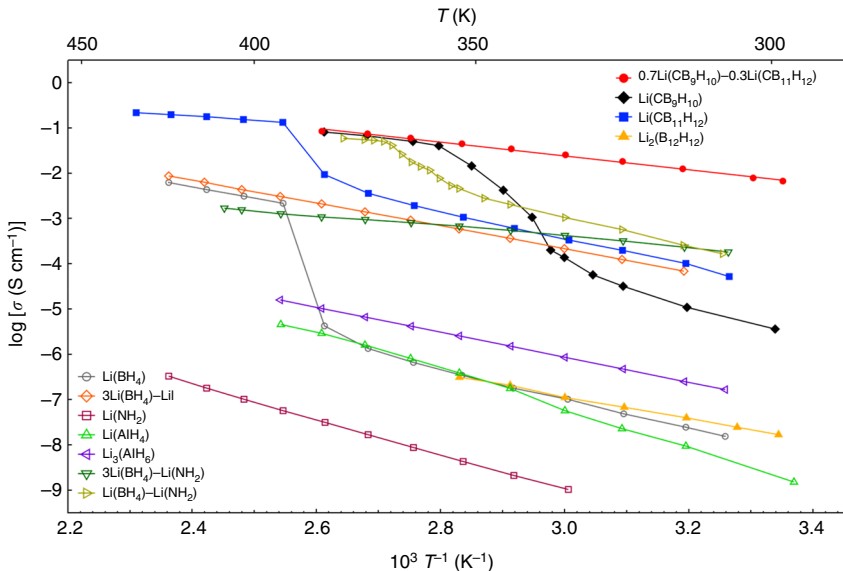

**Fig. 3** Arrhenius plots of the conductivities of $0.7Li(CB_9H_{10})$–$0.3Li(CB_{11}H_{12})$ and those of other complex hydride lithium ion conductors[23,24,41-46]. $0.7Li(CB_9H_{10})$–$0.3Li(CB_{11}H_{12})$ exhibits a high lithium ion conductivity of $6.7 \times 10^{-3} \, S \, cm^{-1}$ at 25 °C, which is the highest value reported for complex hydride lithium ion conductors

compatibility of $0.7Li(CB_9H_{10})$–$0.3Li(CB_{11}H_{12})$ with the lithium metal anode is ascribed to its high chemical stability and high physical deformability.

$0.7Li(CB_9H_{10})$–$0.3Li(CB_{11}H_{12})$ also shows extremely stable lithium ion transfer capability across the interface with the lithium metal anode. When galvanostatically cycled at a current density of $0.2 \, mA \, cm^{-2}$ in both directions for 30 min at 25 °C, the $Li/0.7Li(CB_9H_{10})$–$0.3Li(CB_{11}H_{12})/Li$ cell exhibited very small voltage polarization (6.0 mV) (Fig. 4d). More importantly, the $Li/0.7Li(CB_9H_{10})$–$0.3Li(CB_{11}H_{12})/Li$ cell shows superior voltage retention after repeated lithium plating and stripping cycles. At a current density of $0.2 \, mA \, cm^{-2}$, no voltage changes were observed during 300 cycles (Fig. 4e and Supplementary Fig. 14). The low and unvaried interfacial resistances were also verified by EIS measurements (Supplementary Fig. 15). The superior lithium ion transfer performance confirms the high electrochemical stability of $0.7Li(CB_9H_{10})$–$0.3Li(CB_{11}H_{12})$ against the lithium metal anode.

**All-solid-state lithium metal battery with complex hydride solid electrolyte**. $0.7Li(CB_9H_{10})$–$0.3Li(CB_{11}H_{12})$ can potentially be used to realize a wide range of lithium-metal-based all-solid-state batteries. Based on the high ionic conductivity and high stability with lithium metal of $0.7Li(CB_9H_{10})$–$0.3Li(CB_{11}H_{12})$, the electrochemical investigation was expanded to all-solid-state full cells using $0.7Li(CB_9H_{10})$–$0.3Li(CB_{11}H_{12})$ as the solid electrolyte and lithium metal as the anode. As shown in Fig. 5a, a high-energy-density S electrode (theoretical capacity = $1672 \, mAh \, g^{-1}$; working voltage = 2.1 V vs. $Li^+/Li$) was used as the cathode to take advantage of the high-energy density of the lithium metal anode. The cathode composite was prepared by mixing an S–carbon (C) composite and $0.7Li(CB_9H_{10})$–$0.3Li(CB_{11}H_{12})$ solid electrolyte. The all-solid-state batteries were fabricated by cold-pressing at 153.6 MPa. Detailed cell preparation and measurement conditions are described in the Methods section. Fig. 5b shows the voltage profiles recorded during the first two cycles measured in the voltage range of 1.0–2.5 V (vs. $Li^+/Li$) for a rate of 0.03 C ($50.2 \, mA \, g^{-1}$) at 25 °C. The C-rate is determined as follows: a rate of $nC$ means that the current will (dis)charge the full capacity in $1/n$ hours. In the first cycle, the

discharge capacity and the charge capacity were 2013 and 1557 $mAh \, g^{-1}$, respectively. The higher capacity in the first discharge is presumably attributed to the capacity contribution from the solid electrolyte[38]. After the first discharge, reversible charge–discharge profiles, which present a charge plateau at around 2.2 V and a discharge plateau at around 2.0 V, were observed. The discharge capacity in the second cycle was $1618 \, mAh \, g^{-1}$, which corresponds to 96.8% of the theoretical capacity.

After the first cycle, as the C-rate increased by 1.67, 3.33, 10, and 33.3 times from 0.03 C (1 C = $1672 \, mA \, g^{-1}$), the all-solid-state cell retained 96.5%, 90.0%, 81.4%, and 73.4% of its capacity ($1618 \, mAh \, g^{-1}$) in the second cycle, respectively (Fig. 5c, d). The all-solid-state cells also showed good cycling stability. At 0.1 C (Supplementary Fig. 16 and Supplementary Note 8) and 1 C (Fig. 5e and Supplementary Fig. 17), 98.9% and 84.4% of the capacity (1431 and $1239 \, mAh \, g^{-1}$) in the second cycle was retained after 10 and 20 cycles, respectively. In addition, for a discharging rate of 3 C and a charging rate of 1 C at 50 °C, the discharge capacity was $1533 \, mAh \, g^{-1}$ in the second cycle, and slightly dropped to $1469 \, mAh \, g^{-1}$ after 20 cycles, leading to high-energy densities of $2578–2782 \, Wh \, kg^{-1}$ (Fig. 5f and Supplementary Fig. 18). During cycling, the coulombic efficiencies became saturated at ~100% (Fig. 5e, f, and Supplementary Fig. 16b). From the perspective of energy density, the reversible energy densities of over $2500 \, Wh \, kg^{-1}$ at high current densities of 1–3 C are remarkable, as they are better than those of previously reported Li–S[22,38], Li–LiCoO$_2$ [4,19,36], Li–LiNi$_{0.5}$Mn$_{1.5}$O$_4$[16], and Li–Li$_2$FeMn$_3$O$_8$ [3] all-solid-state batteries.

$0.7Li(CB_9H_{10})$–$0.3Li(CB_{11}H_{12})$ also presents the high stability during prolonged cycling. When cycled for a discharging rate of 5 C and a charging rate of 1 C at 60 °C, the discharge capacity was $1472 \, mAh \, g^{-1}$ in the second cycle, and the reversible capacity of $1017 \, mAh \, g^{-1}$ with the coulombic efficiency of ~100% was retained after 100 cycles (Fig. 6a and Supplementary Fig. 19). An FE-SEM image verifies the preserved intimate contact at the $0.7Li(CB_9H_{10})$–$0.3Li(CB_{11}H_{12})/Li$ interface after cycling (Fig. 6b). Furthermore, no dendrite growth was observed across the $0.7Li(CB_9H_{10})$–$0.3Li(CB_{11}H_{12})/Li$ interface. This physical stability was confirmed in multiple measurement regions (Fig. 6b and

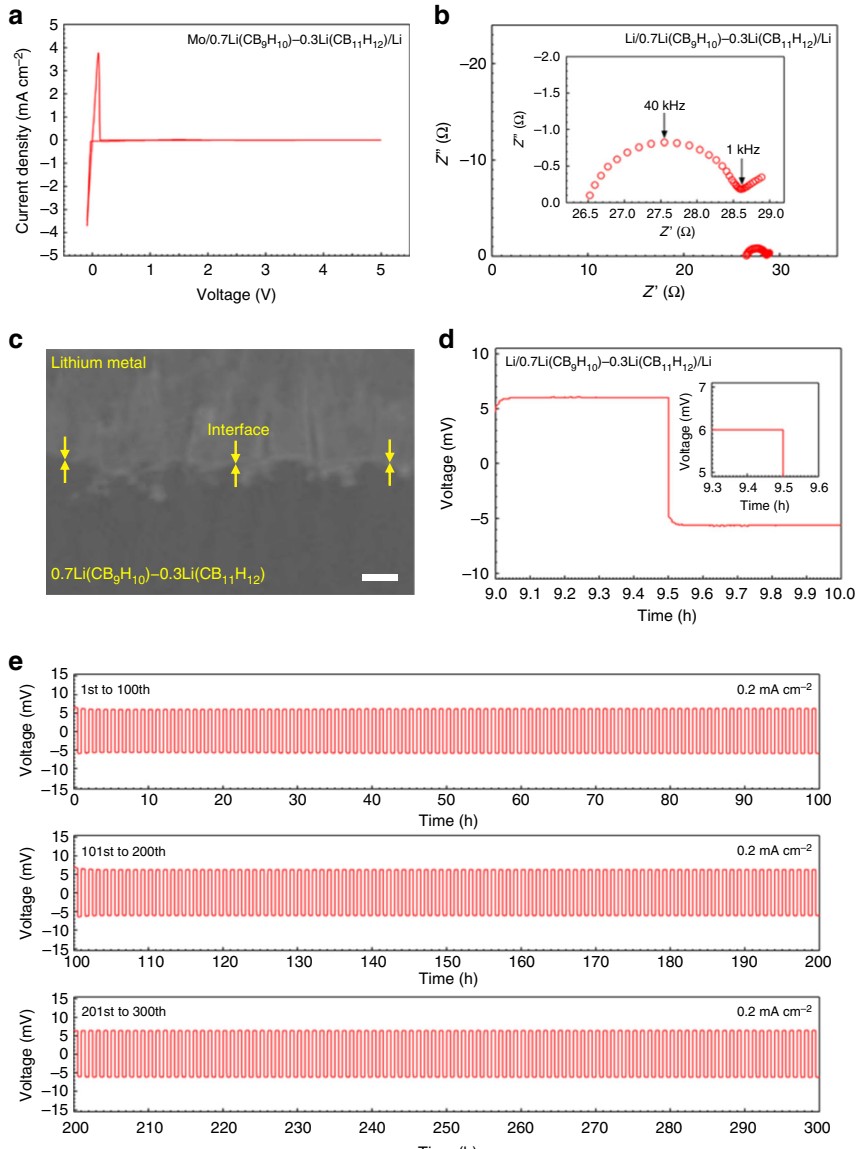

**Fig. 4** Stability of 0.7Li(CB$_9$H$_{10}$)–0.3Li(CB$_{11}$H$_{12}$) with lithium metal. **a** CV curve of a Mo/0.7Li(CB$_9$H$_{10}$)–0.3Li(CB$_{11}$H$_{12}$)/Li cell at a scan rate of 0.5 mV s$^{-1}$ and a scan range of −0.1 to 5.0 V (vs. Li$^+$/Li). **b** Nyquist plot of a Li/0.7Li(CB$_9$H$_{10}$)–0.3Li(CB$_{11}$H$_{12}$)/Li cell. Inset shows the magnified plot in the region of the semicircle indicating the 0.7Li(CB$_9$H$_{10}$)–0.3Li(CB$_{11}$H$_{12}$)/Li interfacial resistance. **c** FE-SEM image of the 0.7Li(CB$_9$H$_{10}$)–0.3Li(CB$_{11}$H$_{12}$)/Li interface. **d** 10th galvanostatic cycling profile of the Li/0.7Li(CB$_9$H$_{10}$)–0.3Li(CB$_{11}$H$_{12}$)/Li cell. Inset shows the magnified profile. **e** Galvanostatic cycling profiles for prolonged cycles. 1st–100th cycles (top), 101st–200th cycles (middle), and 201st–300th cycles (bottom). All electrochemical measurements were conducted at 25 °C. Scale bar, 30 μm in **c**

Supplementary Fig. 20). In addition, XRD (Fig. 6c) and Raman spectroscopy (Fig. 6d) measurements indicate that the characteristic peaks of 0.7Li(CB$_9$H$_{10}$)–0.3Li(CB$_{11}$H$_{12}$) are maintained after cycling. These results demonstrate the practical feasibility of the 0.7Li(CB$_9$H$_{10}$)–0.3Li(CB$_{11}$H$_{12}$) solid electrolyte for all-solid-state batteries employing the lithium metal anode.

## Discussion

We developed the complex hydride lithium superionic conductor 0.7Li(CB$_9$H$_{10}$)–0.3Li(CB$_{11}$H$_{12}$). The partial incorporation of Li(CB$_{11}$H$_{12}$) into Li(CB$_9$H$_{10}$) allows the disordered high-$T$ phase of Li(CB$_9$H$_{10}$) to stabilize at room temperature. As a result of the effects of the disordered complex hydride phase, a room-temperature lithium superionic conductivity of over 10$^{-3}$ S cm$^{-1}$ was achieved for the first time for a pure complex hydride. Of note,

this study provides a general guideline for how to develop lithium superionic conductors based on *closo*-type complex hydrides: (1) find a host *closo*-type complex hydride that exhibits a high ionic conductivity in the disordered high-$T$ phase, (2) find a *closo*-type complex anion whose structure and size are similar to those of the host material, and (3) partially incorporate the given complex anion into the host material to form a disordered structure, which stabilizes the high-$T$ phase at lower temperatures.

The synthesis principle used in this study is different from those used previously for *closo*-type lithium ion conductors with multiple complex anions. Since previous approaches focused on synthetic methods without considerations of material compositions, the produced compounds contained a massive amount of residual products[39,40] due to incomplete mixing of complex anions, complicating efforts to apply the conductors to batteries, as well as making precise analyses of conduction properties difficult.

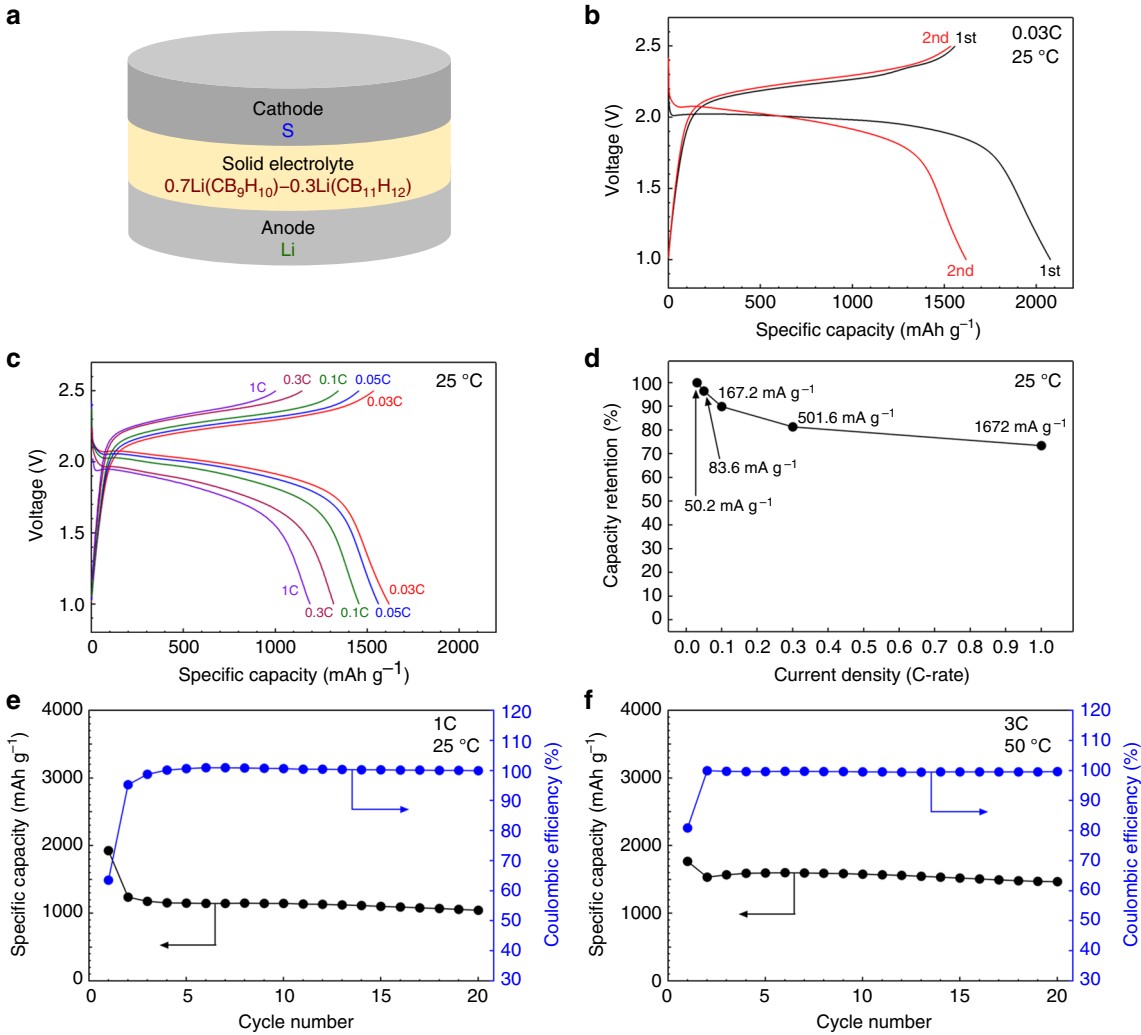

**Fig. 5** High-energy-density all-solid-state lithium metal batteries. **a** Schematic illustration of the prepared all-solid-state batteries. S, Li/0.7Li(CB$_9$H$_{10}$)–0.3Li (CB$_{11}$H$_{12}$), and lithium metal were used as the cathodes, solid electrolytes, and anodes, respectively. **b** Voltage profiles for a rate of 0.03 C (50.2 mA g$^{-1}$) at 25 °C during the first two cycles. **c** Discharge–charge profiles at 0.03, 0.05, 0.1, 0.3, and 1 C after an initial cycle at 25 °C. **d** Capacity retention as a function of current density. **e**, **f** Cycling performances of discharge capacity and coulombic efficiency (**e**) for a rate of 1 C at 25 °C and (**f**) for a discharging rate of 3 C and a charging rate of 1 C at 50 °C

The developed *closo*-type complex hydride has various advantages as a solid electrolyte in all-solid-state batteries. Its intrinsic stability with the lithium metal anode enables facile and stable lithium ion transfer at the interface. The high ionic conductivity of 0.7Li(CB$_9$H$_{10}$)–0.3Li(CB$_{11}$H$_{12}$) coupled with its stability with lithium metal enables excellent performance of high-energy-density Li–S batteries in a wide temperature range. From the battery manufacturing perspective, the superior deformability of 0.7Li(CB$_9$H$_{10}$)–0.3Li(CB$_{11}$H$_{12}$) facilitates the preparation of compact solid electrolytes and electrode/electrolyte interface, resulting in intimate contact throughout the battery. The unique properties of the developed complex hydride solid electrolyte will not only inspire future efforts to find lithium superionic conductors based on complex hydrides, but also opens up a new group of solid electrolytes for practical all-solid-state lithium metal batteries that may lead to the development of high-energy-density electrochemical devices.

## Methods
**Synthesis**. The starting materials, Li(CB$_9$H$_{10}$) and Li(CB$_{11}$H$_{12}$), were obtained by drying Li(CB$_9$H$_{10}$)·$x$H$_2$O (Katchem) at 200 °C for 12 h and Li(CB$_{11}$H$_{12}$)·$x$H$_2$O (Katchem) at 160 °C for 12 h under vacuum (<5 × 10$^{-4}$ Pa). Li(CB$_9$H$_{10}$) and Li (CB$_{11}$H$_{12}$) were weighted in the appropriate molar ratios and ground using a mortar and pestle for 15 min. To synthesize the $(1-x)$(CB$_9$H$_{10}$)–$x$0.3Li(CB$_{11}$H$_{12}$) ($x$ = 0.1, 0.3, and 0.5) complex hydrides, the mixed powder was mechanically milled using a planetary ball mill (Pulverisette 7, Fritsch) with 20 steel balls (each 7 mm in diameter) at 400 rpm for 20 h. After ball-milling, the powders were reground using a mortar and pestle for 15 min. All of the procedures were conducted under an Ar atmosphere.

**Characterization**. The phase analysis was performed with XRD (X'PERT Pro, PANalytical) measurements with CuKα radiation (wavelength $\lambda$ = 1.5406 Å for Kα1 and 1.5444 Å for Kα2) in the 2θ range of 7–50°. The powder for the XRD measurements was loaded into a thin-walled glass capillary under an Ar atmosphere and sealed with paraffin liquid. DTA was performed using a Rigaku Thermo Plus TG-8120 system from 25 to 200 °C at a ramping rate of 5 °C min$^{-1}$ under an Ar flow (150 cc min$^{-1}$). The vibrational modes of complex anions were characterized by Raman spectroscopy (DXR, Thermo Scientific). Morphologies and particle sizes were analyzed using FE-SEM (SU9000, Hitachi).

**Ionic conductivity measurement**. Fifty milligrams of the powder was first placed into a 10-mm-diameter Teflon guide and uniaxially pressed at 153.6 MPa to prepare the pellet-type sample (0.10 cm in thickness and 0.785 cm$^2$ in surface area). Subsequently, Au electrode powders were transferred onto both sides of the pressed sample still present in the Teflon guide and uniaxially pressed again at 153.6 MPa to obtain a single pellet comprising the solid electrolyte and Au electrodes. No additional sintering process was conducted to reduce grain boundaries before measurements. Ionic conductivities were measured using the AC impedance

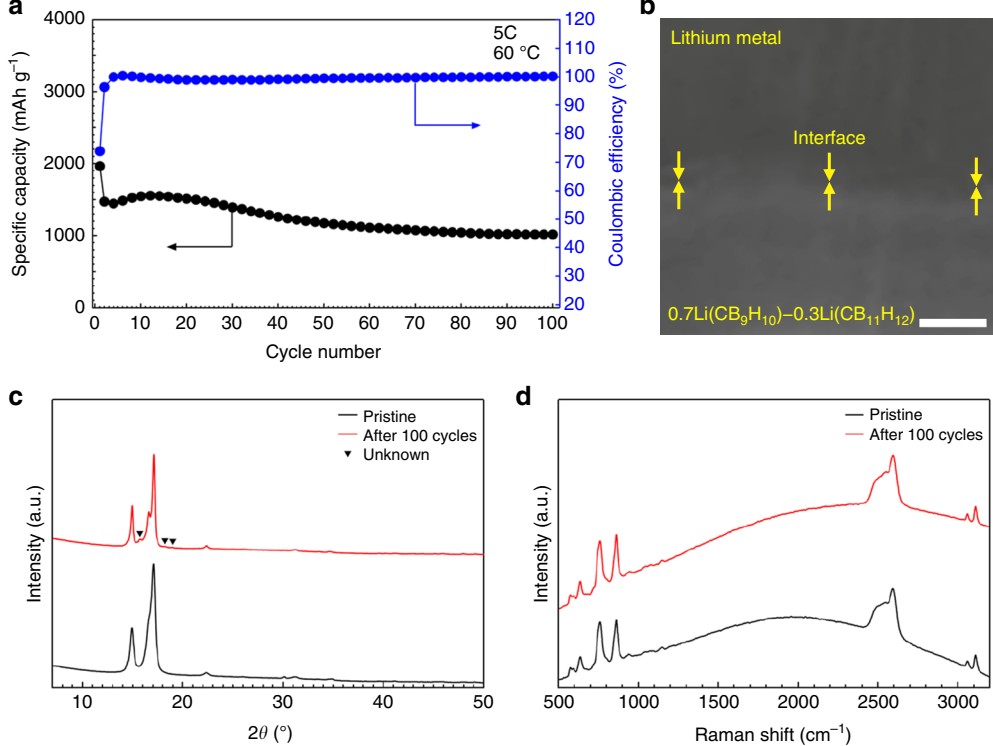

**Fig. 6** Stability of 0.7Li(CB$_9$H$_{10}$)–0.3Li(CB$_{11}$H$_{12}$) during prolonged cycling. **a** Cycling performance of discharge capacity and coulombic efficiency for a discharging rate of 5 C and a charging rate of 1 C at 60 °C. **b** FE-SEM image of the 0.7Li(CB$_9$H$_{10}$)–0.3Li(CB$_{11}$H$_{12}$)/Li interface after 100 cycles. **c** XRD patterns and **d** Raman profiles of 0.7Li(CB$_9$H$_{10}$)–0.3Li(CB$_{11}$H$_{12}$) before and after 100 cycles. A small amount of an unknown phase was detected in the XRD pattern. Scale bar, 20 μm in **b**

method over a temperature range of 25–110 °C K with applied frequencies of 4 Hz–1 MHz using a frequency response analyzer (3532-80, HIOKI). Lithium self-diffusion coefficients were estimated using the PFG stimulated-echo method of $^7$Li NMR with a Bruker Avance 400 NMR spectrometer equipped with a Bruker Diff60 probe operating at a resonance frequency of 155.6 MHz.

**Stability evaluation**. The electrochemical stability was evaluated by CV (1470E, Solartron Analytical) using a Mo/0.7Li(CB$_9$H$_{10}$)–0.3Li(CB$_{11}$H$_{12}$)/Li cell in a scan range of −0.1 to 5 V (vs. Li$^+$/Li). The interfacial resistance between 0.7Li (CB$_9$H$_{10}$)–0.3Li(CB$_{11}$H$_{12}$) and lithium metal was examined with a frequency response analyzer (3532-80, Hioki) using a Li/0.7Li(CB$_9$H$_{10}$)–0.3Li(CB$_{11}$H$_{12}$)/Li cell. Lithium plating/stripping cycling was conducted with a Li/0.7Li (CB$_9$H$_{10}$)–0.3Li(CB$_{11}$H$_{12}$)/Li cell at a current density of 0.2 mA cm$^{-2}$ using a battery tester (580 Battery Test System, Scribner Associates). For the symmetric Li/0.7Li(CB$_9$H$_{10}$)–0.3Li(CB$_{11}$H$_{12}$)/Li cell, a 0.7Li(CB$_9$H$_{10}$)–0.3Li(CB$_{11}$H$_{12}$) pellet with a larger thickness (0.14 cm) than that (0.10 cm) used for the conductivity evaluation was prepared to avoid lithium–lithium shorting. Mo foil (Nilaco) and lithium foil (Honjo Metal) were used as electrodes of the Mo/0.7Li(CB$_9$H$_{10}$)–0.3Li (CB$_{11}$H$_{12}$)/Li and Li/0.7Li(CB$_9$H$_{10}$)–0.3Li(CB$_{11}$H$_{12}$)/Li cells. For cell fabrication, 50–70 mg of the 0.7Li(CB$_9$H$_{10}$)–0.3Li(CB$_{11}$H$_{12}$) powder was placed into a 10-mm-diameter Teflon guide and uniaxially pressed at 38.4 MPa. The electrodes were transferred onto the pressed sample still present in the Teflon guide and uniaxially pressed again at 153.6 MPa to prepare pellet-type cells. The cross-section of the 0.7Li(CB$_9$H$_{10}$)–0.3Li(CB$_{11}$H$_{12}$)/Li interface was observed using FE-SEM (S-3400N, Hitachi). To prevent lithium metal from sticking to the cell, Cu foil was inserted between the lithium metal and cell. The FE-SEM specimen for the cross-section observation was prepared using a dual-beam focused ion beam (FIB, FB2200, Hitachi).

**Battery assembly and electrochemical test**. All of the cell preparation processes were conducted under an Ar atmosphere inside a glove box. Discharge–charge experiments were performed with a S/0.7Li(CB$_9$H$_{10}$)–0.3Li(CB$_{11}$H$_{12}$)/Li cell. For cathode composite fabrication, an S–C composite was first prepared. Elemental S (Sigma-Aldrich), Ketjen black (KB, ECP600JD, Lion Corp.), and Maxsorb (MSC-30, Kansai Coke and Chemicals Co., Ltd) were weighted in a 50:25:25 (wt %) ratio and ball-milled at 400 rpm for 20 h under an Ar atmosphere[22]. The S–C composite and 0.7Li(CB$_9$H$_{10}$)–0.3Li(CB$_{11}$H$_{12}$) powders were then mixed at a 1:1 mass ratio using an agate mortar and pestle for 15 min and used as the cathode composite. For cell fabrication, 50–70 mg of the 0.7Li(CB$_9$H$_{10}$)–0.3Li(CB$_{11}$H$_{12}$) powder was placed into a 10-mm-diameter Teflon guide and uniaxially pressed

at 38.4 MPa. Subsequently, ~8 mg of the cathode composite powder and lithium foil anode were transferred onto opposite sides of the electrolyte still present in the Teflon guide and uniaxially pressed again at 153.6 MPa to prepare pellet-type cells. The cells were cycled in the voltage range of 1.0–2.5 V (vs. Li$^+$/Li) using a battery tester (580 Battery Test System, Scribner Associates). The C-rate in this study is defined based on 1 C = 1672 mA g$^{-1}$. For the calculation of the gravimetric capacities and currents, only the mass of the S cathode material was taken into account. The specific energy density was calculated using

$$E = \frac{1}{m}\int_0^T i \times V\, dt \qquad (1)$$

where $i$ is the static current, $V$ is the cell voltage, $T$ is the discharging time required to reach the cutoff voltage, and $m$ is the mass of the S cathode material.

## Data availability

The data that support the findings of this study are available from the corresponding authors upon reasonable request.

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

## Acknowledgements

This work was supported by JSPS KAKENHI (Grant–in–Aid for Research Activity Start–up, 17H06519 and Grants–in–Aid for Scientific Research on Innovative Areas "Hydrogenomics", JP18H05513 and JP18H05518), the Collaborative Research Center on Energy Materials in IMR (E–IMR), and Advanced Target Project–4 of WPI–AIMR, Tohoku University. The authors would like to thank H. Ohmiya and N. Warifune for technical assistance (Tohoku University).

## Author contributions

S.K. designed the research. S.K. carried out the syntheses, measurements, and data analyses. H.O. assisted in the electrochemical measurements. N.T. assisted in the Raman spectroscopy measurements. T.S. assisted in the phase analyses using XRD patterns. S.T. and T.O. participated in discussions of the results. D.A., N.K. and J.K. performed the NMR measurements. S.K. wrote the manuscript. S.K. and S.O. supervised the research. All authors discussed the results and commented on the manuscript.

## Additional information

**Competing interests:** The authors declare no competing interests.

