## [Peer Review File · Nature Communications]

Reviewers' comments:

Reviewer #2 (Remarks to the Author):

The manuscript presents a highly conducting solid electrolyte, which is a solid solution of Li(CB9H10) and Li(CB11H12), and all-solid-state Li metal batteries made of that electrolyte. The conductivity value at room temperature is impressive and the electrochemical cell performance certainly outperforms other hydride Li ion conductors. However, almost same kind of electrolyte was previously reported, which makes the current work less of a novelty. In this regard, the composition and the cycle test of the electrochemical cell, which is firstly presented here, is noteworthy, but more extensive cycle test is recommended to prove the utility of such batteries in real application. Several questions and comments are as follows.

1. The design strategy for the same type of electrolyte was already published by Tang et al. (ref. 37). The major advancement from the previous publication is the realization of the phase-pure high-T phase down to room temperature by changing the mixing ratio from $x=0.5$ to $x=0.3$. In ref. 37, the mixture had been mostly composed of the high-T hexagonal phase after ball-milling, but the cubic phase appeared after annealing at 473K. It is plausible that $x=0.3$ would undergo the same phase separation although the solid solution appears stable under the DTA experiment condition. It is recommended to test the thermodynamic stability of the solid solution by subjecting it to an additional heat treatment.
2. In the FE-SEM image in Fig. 4c, what are the two bright lines emerging from the interface stretched to the bottom in the electrolyte region? Is the cold-pressed electrolyte dense enough to suppress Li penetration during cycling?
3. Considering that the solid electrolyte itself is not completely new, the performance of the electrochemical cell with Li metal anode constitutes the main part of the current manuscript. While the stability upon repeated lithium plating/stripping up to 300 cycles was demonstrated, discharge/charge experiment results were shown only up to 20 cycles. The cycle number is rather short to prove the utility of such all-solid-state lithium metal batteries. It is recommended to present the cycle test result over 100 cycle numbers. In addition, after the prolonged cycle test, it would be informative to show i) the stability of the electrolyte using XRD and Raman spectra and ii) the image of the Li/electrolyte interface.

Reviewer #3 (Remarks to the Author):

In their manuscript, Kim et al. demonstrate the solid-state synthesis of a solid-solution closo-borate ionic conductor with excellent room-temperature Li⁺ conductivity and stability against Li metal. This new class of materials is highly promising, and the authors have demonstrated a significant advance in overcoming the instability of the disordered phase at lower temperatures, which has been a primary shortcoming. Equally significantly, the authors were able to show that the material demonstrates exceedingly low interfacial resistance, along with reversible Li plating/stripping. They have also demonstrated a working Li-S cell, helping to demonstrate the practical viability of closo-borates for actual devices.

Overall, this is an impressive study with relevance to a broad readership. The paper is clearly written, and the conclusions are well supported by the available data. I recommend that the paper be published in Nature Communications. I have a few minor questions/suggestions:

- 1) The significantly broadened XRD peaks suggest that the crystallite size may be quite small (also mentioned by the authors in the Supplementary Material). Can the authors speculate about the possible implications of these small grains on the promotion of disorder and high ionic

conductivity? Although I agree that the two-phase coexistence observed in Ref. 37 appears to be absent here, is it possible that the elevated fraction of grain boundaries could be aiding the conductivity by introducing additional interfacial disorder? Alternatively, could the small grain sizes be stabilizing the solid solution behavior, thereby indirectly aiding ion diffusion? To gauge these possibilities, it may be worthwhile to extract the crystallite/grain size from the XRD peak widths in order to estimate of the fraction of material associated with grain boundaries. As the authors state on p. 7, it appears to be difficult to distinguish between bulk and grain boundary resistances for the mixed phase based only on EIS.

2) Although the compositions differ, in addition to Ref. 37, the authors should also reference other successful attempts to create room-temperature mixed-closo-borate solid solution electrolytes (e.g., DOI: 10.1039/c7cc00794a, 10.1039/c7ee02420g, 10.1016/j.jpowsour.2018.09.085). This will better contextualize the authors' results.

3) Please label a couple of frequencies on the Nyquist plots in Fig. 2 to orient the reader (following what was done in Fig. 4b).

4) p. 8: Beyond Refs. 24 and 26 (and Refs. 27-29), there have been recent theoretical studies that have investigated the mechanisms of ionic conductivity and cooperativity more specifically in disordered closo-borate systems (DOI: 10.1039/C6TA07443J, 10.1002/aenm.201703422, 10.1021/acs.chemmater.7b02902). It may be helpful to refer to these in the discussion of possible motivations for fast ion conduction in the $0.7\text{Li}(\text{CB9H10})-0.3\text{Li}(\text{CB11H12})$ system.

Responses to Reviewers' Comments

Reviewer #2

The manuscript presents a highly conducting solid electrolyte, which is a solid solution of $\text{Li}(\text{CB}_9\text{H}_{10})$ and $\text{Li}(\text{CB}_{11}\text{H}_{12})$, and all-solid-state Li metal batteries made of that electrolyte. The conductivity value at room temperature is impressive and the electrochemical cell performance certainly outperforms other hydride Li ion conductors. However, almost same kind of electrolyte was previously reported, which makes the current work less of a novelty. In this regard, the composition and the cycle test of the electrochemical cell, which is firstly presented here, is noteworthy, but more extensive cycle test is recommended to prove the utility of such batteries in real application. Several questions and comments are as follows.

Response: We are grateful to the reviewer for his/her comprehensive review of our manuscript and encouraging comments.

1. The design strategy for the same type of electrolyte was already published by Tang et al. (ref. 37). The major advancement from the previous publication is the realization of the phase-pure high-T phase down to room temperature by changing the mixing ratio from $x=0.5$ to $x=0.3$. In ref. 37, the mixture had been mostly composed of the high-T hexagonal phase after ball-milling, but the cubic phase appeared after annealing at 473K. It is plausible that $x=0.3$ would undergo the same phase separation although the solid solution appears stable under the DTA experiment condition. It is recommended to test the thermodynamic stability of the solid solution by subjecting it to an additional heat treatment.

Response: We found this reviewer's point extremely helpful in improving the quality of the manuscript. This is much appreciated. According to the reviewer's suggestion, the thermodynamic stability of $0.7\text{Li}(\text{CB}_9\text{H}_{10})-0.3\text{Li}(\text{CB}_{11}\text{H}_{12})$ was assessed after additional heat treatment at 473 K for 12 h as shown below. The XRD measurements clarify that the high-temperature phase of $0.7\text{Li}(\text{CB}_9\text{H}_{10})-0.3\text{Li}(\text{CB}_{11}\text{H}_{12})$ is preserved after this heat treatment. These results confirm the high thermodynamic stability of $0.7\text{Li}(\text{CB}_9\text{H}_{10})-0.3\text{Li}(\text{CB}_{11}\text{H}_{12})$. To reflect this point, XRD data obtained before and after the heat treatment were added to Fig. S8 and the following sentence was added to the revised manuscript:

[Page 6]

...high- T phase at room temperature. Additionally, the high thermal stability of $0.7\text{Li}(\text{CB}_9\text{H}_{10})-0.3\text{Li}(\text{CB}_{11}\text{H}_{12})$ was verified by XRD measurements after the heat-treatment at 473 K for 12 h (Supplementary Fig. 8).

[Fig. S8 in SI]

Supplementary Figure 8. Thermal stability of $0.7\text{Li}(\text{CB}_9\text{H}_{10})-0.3\text{Li}(\text{CB}_{11}\text{H}_{12})$. XRD profiles of $0.7\text{Li}(\text{CB}_9\text{H}_{10})-0.3\text{Li}(\text{CB}_{11}\text{H}_{12})$ before and after heat-treatment at 473 K for 12 h.

2. In the FE-SEM image in Fig. 4c, what are the two bright lines emerging from the interface stretched to the bottom in the electrolyte region? Is the cold-pressed electrolyte dense enough to suppress Li penetration during cycling?

Response: In the original manuscript, the solid electrolyte/lithium metal pellet for the FE-SEM measurements was obtained by peeling it off the cell with tweezers, as the lithium metal has high adhesive strength. This process is likely to apply considerable tension to the solid electrolyte/lithium metal interface, which can lead to sample damage and associated cleavage formation, shown as bright lines in Fig. 4c. To overcome this problem, Cu foil was inserted between the solid electrolyte/lithium metal pellet and the cell. This method allowed us to take out the pellet from the cell without damage. The FE-SEM measurement conducted with this new sample showed no cleavages, even over a wider measurement region than in the previous experiment. This indicates the high density of the prepared solid electrolyte, which can suppress Li penetration during cycling. Indeed, no dendrite growth was observed in multiple measurement regions even after prolonged discharge-charge reactions over 100 cycles. To illustrate these points, the new FE-SEM results before and after cycling were added to Figs 4c, 6b and S20, and the following sentences were added to the revised manuscript:

[Page 12]

...Supplementary Fig. 19). An FE-SEM image confirms the preserved intimate contact at the $0.7\text{Li}(\text{CB}_9\text{H}_{10})-0.3\text{Li}(\text{CB}_{11}\text{H}_{12})/\text{Li}$ interface after cycling (Fig. 6b). Furthermore, no dendrite growth was observed across the $0.7\text{Li}(\text{CB}_9\text{H}_{10})-0.3\text{Li}(\text{CB}_{11}\text{H}_{12})/\text{Li}$ interface. This physical stability was confirmed in multiple measurement regions (Fig. 6b and Supplementary Fig. 20).

[Fig. 6b]

[Fig. S20 in SI]

Supplementary Figure 20. Stability of the $0.7\text{Li}(\text{CB}_9\text{H}_{10})-0.3\text{Li}(\text{CB}_{11}\text{H}_{12})/\text{Li}$ interface after cycling. **a,b,** FE-SEM images of the $0.7\text{Li}(\text{CB}_9\text{H}_{10})-0.3\text{Li}(\text{CB}_{11}\text{H}_{12})/\text{Li}$ interface for multiple measurement regions after 100 cycles.

[Page 15]

...(S-3400 N, Hitachi). To prevent lithium metal from sticking to the cell, Cu foil was inserted between the lithium metal and cell.

[Fig. 4c]

3. Considering that the solid electrolyte itself is not completely new, the performance of the electrochemical cell with Li metal anode constitutes the main part of the current manuscript. While the stability upon repeated lithium plating/stripping up to 300 cycles was demonstrated, discharge/charge experiment results were shown only up to 20 cycles. The cycle number is rather short to prove the utility of such all-solid-state lithium metal batteries. It is recommended to present the cycle test result over 100 cycle numbers. In addition, after the prolonged cycle test, it would be informative to show i) the stability of the electrolyte using XRD and Raman spectra and ii) the image of the Li/electrolyte interface.

Response: We appreciate the reviewer's critical point. Following the reviewer's suggestion, the discharge-charge experiment was performed for 100 cycles. To improve the cyclability, a cycle test was conducted at a higher temperature (60°C) than for the previous 20-cycle experiment (50°C). The discharging and charging rates were 5C and 1C, respectively. The discharge capacity was 1,472 mAh g⁻¹ in the second cycle, and a reversible capacity of 1,017 mAh g⁻¹ with a coulombic efficiency of ~100% was retained after 100 cycles. Furthermore, the stability of the solid electrolyte was investigated further by ex-situ SEM, XRD and Raman measurements after cycling. An SEM image verified close physical contact at the interface between 0.7Li(CB₉H₁₀)-0.3Li(CB₁₁H₁₂) and lithium metal after cycling. Especially, as described in Response 2, no dendrite growth was observed at multiple measurement regions (Figs. 6b and S20). Additionally, although a small amount of an unknown phase was detected from the XRD measurement, characteristic peaks of 0.7Li(CB₉H₁₀)-0.3Li(CB₁₁H₁₂) were clearly observed in the XRD and Raman spectra. These results confirm the high stability of 0.7Li(CB₉H₁₀)-0.3Li(CB₁₁H₁₂) during discharge-charge reactions. To illustrate these points, discharge-charge results over 100 cycles, and SEM, XRD, and Raman results were added to Figs. 6, S19 and S20, and the following sentences were added to the revised manuscript:

[Page 12]

0.7Li(CB₉H₁₀)-0.3Li(CB₁₁H₁₂) also presents high stability during prolonged cycling. When cycled for a discharging rate of 5C and a charging rate of 1C at 60°C, the discharge capacity was 1472 mAh g⁻¹ in the second cycle, and a reversible capacity of 1017 mAh g⁻¹ with a coulombic efficiency of ~100% was retained after 100 cycles (Fig. 6a and Supplementary Fig. 19). An FE-SEM image confirms the preserved intimate contact at the 0.7Li(CB₉H₁₀)-0.3Li(CB₁₁H₁₂)/Li interface after cycling (Fig. 6b). Furthermore, no dendrite growth was observed across the 0.7Li(CB₉H₁₀)-0.3Li(CB₁₁H₁₂)/Li interface. This physical stability was confirmed in multiple measurement regions (Fig. 6b and Supplementary Fig. 20). Additionally, XRD (Fig. 6c) and Raman spectroscopy (Fig. 6d) measurements indicate that the characteristic peaks of 0.7Li(CB₉H₁₀)-0.3Li(CB₁₁H₁₂) were maintained after cycling.

[Fig. 6]

Figure 6 | Stability of $0.7\text{Li}(\text{CB}_9\text{H}_{10})-0.3\text{Li}(\text{CB}_{11}\text{H}_{12})$ during prolonged cycling. **a**, Cycling performance of discharge capacity and coulombic efficiency for a discharging rate of 5C and a charging rate of 1C at 60°C. **b**, FE-SEM image of the $0.7\text{Li}(\text{CB}_9\text{H}_{10})-0.3\text{Li}(\text{CB}_{11}\text{H}_{12})/\text{Li}$ interface after 100 cycles. **c,d**, (c) XRD patterns and (d) Raman profiles of $0.7\text{Li}(\text{CB}_9\text{H}_{10})-0.3\text{Li}(\text{CB}_{11}\text{H}_{12})$ before and after 100 cycles. A small amount of an unknown phase was detected in the XRD pattern.

[Fig. S19 in SI]

Supplementary Figure 19. Discharge-charge profiles for prolonged cycles. Discharge-charge profiles of a $\text{S}/0.7\text{Li}(\text{CB}_9\text{H}_{10})-0.3\text{Li}(\text{CB}_{11}\text{H}_{12})/\text{Li}$ cell for a discharging rate of 5C and a charging rate of 1C at 60°C.

[Fig. S20 in SI]

Supplementary Figure 20. Stability of the $0.7\text{Li}(\text{CB}_9\text{H}_{10})-0.3\text{Li}(\text{CB}_{11}\text{H}_{12})/\text{Li}$ interface after cycling. a,b, FE-SEM images of the $0.7\text{Li}(\text{CB}_9\text{H}_{10})-0.3\text{Li}(\text{CB}_{11}\text{H}_{12})/\text{Li}$ interface for multiple measurement regions after 100 cycles.

Reviewer #3

In their manuscript, Kim et al. demonstrate the solid-state synthesis of a solid-solution closo-borate ionic conductor with excellent room-temperature Li⁺ conductivity and stability against Li metal. This new class of materials is highly promising, and the authors have demonstrated a significant advance in overcoming the instability of the disordered phase at lower temperatures, which has been a primary shortcoming. Equally significantly, the authors were able to show that the material demonstrates exceedingly low interfacial resistance, along with reversible Li plating/stripping. They have also demonstrated a working Li-S cell, helping to demonstrate the practical viability of closo-borates for actual devices.

Overall, this is an impressive study with relevance to a broad readership. The paper is clearly written, and the conclusions are well supported by the available data. I recommend that the paper be published in Nature Communications. I have a few minor questions/suggestions:

Response: We are grateful to the reviewer for his/her comprehensive review of our manuscript and constructive comments.

1. The significantly broadened XRD peaks suggest that the crystallite size may be quite small (also mentioned by the authors in the Supplementary Material). Can the authors speculate about the possible implications of these small grains on the promotion of disorder and high ionic conductivity? Although I agree that the two-phase coexistence observed in Ref. 37 appears to be absent here, is it possible that the elevated fraction of grain boundaries could be aiding the conductivity by introducing additional interfacial disorder? Alternatively, could the small grain sizes be stabilizing the solid solution behavior, thereby indirectly aiding ion diffusion? To gauge these possibilities, it may be worthwhile to extract the crystallite/grain size from the XRD peak widths in order to estimate of the fraction of material associated with grain boundaries. As the authors state on p. 7, it appears to be difficult to distinguish between bulk and grain boundary resistances for the mixed phase based only on EIS.

Response: We found this reviewer's point extremely helpful in improving the quality of the manuscript. This is much appreciated. According to the reviewer's suggestion, the grain size was estimated using the XRD peak widths. The average grain size of 0.7Li(CB₉H₁₀)–0.3Li(CB₁₁H₁₂) was ~100 μm. To experimentally verify this size effect, only the starting material (LiCB₉H₁₀) was ball-milled, and its ionic conductivity was measured as shown below. The grain size of ball-milled LiCB₉H₁₀ was ~100 μm, which is consistent with that of 0.7Li(CB₉H₁₀)–0.3Li(CB₁₁H₁₂). The impedance results indicate that ball-milled LiCB₉H₁₀ shows no change in conductivity compared to pristine LiCB₉H₁₀. These results confirm that the size effect on the disordering and/or conductivity is negligible. To reflect this point, the impedance results for ball-milled LiCB₉H₁₀ and the following phrase were added to the revised manuscript:

[Fig. S10 in SI]

Supplementary Figure 10. Effect of ball-milling on the conductivity. Ionic conductivities of Li(CB₉H₁₀) before and after ball-milling at 400 rpm for 20 h.

The average grain size of $\text{LiCB}_9\text{H}_{10}$ after ball-milling, which was estimated from the XRD peak widths in Supplementary Fig. 7a, was $\sim 100 \mu\text{m}$. The impedance results indicate that the size has a negligible effect on the conductivity.

[Page 8]

...high- T phase (Fig. 2b and Supplementary Fig. 10).

2. Although the compositions differ, in addition to Ref. 37, the authors should also reference other successful attempts to create room-temperature mixed-closo-borate solid solution electrolytes (e.g., DOI: 10.1039/c7cc00794a, 10.1039/c7ee02420g, 10.1016/j.jpowsour.2018.09.085). This will better contextualize the authors' results.

Response: We appreciate the reviewer's constructive suggestion. The above references were added to the revised manuscript as follows:

[Page S22]

Various *closo*-type complex hydrides have been investigated for sodium ion conductors¹¹⁻¹³.

3. Please label a couple of frequencies on the Nyquist plots in Fig. 2 to orient the reader (following what was done in Fig. 4b).

Response: Following the reviewer's suggestion, three frequencies were added to Fig. 2a as shown below:

[Fig. 2a]

4. p. 8: Beyond Refs. 24 and 26 (and Refs. 27-29), there have been recent theoretical studies that have investigated the mechanisms of ionic conductivity and cooperativity more specifically in disordered closo-borate systems (DOI: 10.1039/C6TA07443J, 10.1002/aenm.201703422, 10.1021/acs.chemmater.7b02902). It may be helpful to refer to these in the discussion of possible motivations for fast ion conduction in the $0.7\text{Li}(\text{CB}_9\text{H}_{10})-0.3\text{Li}(\text{CB}_{11}\text{H}_{12})$ system.

Response: Once again, we appreciate the reviewer's constructive suggestion. The above references were added to the revised manuscript as follows:

[Page 8]

...respectively (Supplementary Table 1). It has been reported that the fast ionic conduction in the high- T phase of complex hydrides results from their vacancy-rich disordered cation sublattices within networks of reorientationally disordered complex anions^{24,26-29}.

Reviewers' comments:

Reviewer #1 (Remarks to the Author):

The authors have conducted a study on a highly conducting solid electrolyte, solid solution of Li(CB9H10) and Li(CB11H12), and present valuable results on its application in solid-state lithium-sulfur batteries. With this electrolyte, they have obtained very high ionic conductivities at room temperature, together with a very good stability against lithium and outstanding cell performances. As opposed to already published study, on almost the same type of electrolyte, the new proposed stoichiometry ensure the high temperature phase stability at room temperature. The number of cycle test, 100, albeit at 60 C but at rather high charge / discharge rates, is very encouraging for practical application of the batteries.

In the revised version of the manuscript, the authors have taken careful care of the suggestions and comments from the reviewers. In their rebuttal, they have properly and soundly address the different remarks and questions.

The only weak point may concern their evaluation of the grain size for the ball-milled powder. The author do not explain how they actually evaluated the grain size. If, it is based on the Scherrer equation, then the value given is certainly wrong as it is commonly accepted that this equation is valid for crystallites with sizes up to 1 micrometer at max. I recommend the authors, either to prove the validity of the reported value or to delete it from the manuscript and supplementary information. Their addition concerning the conductivity measurement of the ball milled Li(CB9H10) appears to be sufficient to answer the reviewer's #3 question 1). More interestingly, from Figure 8, one can see the sharpening of the peaks after the heat treatment. It may be related to some annealing of the defect caused by the ball milling. If the authors could provide a conductivity measurement for this sample, this will certainly close the case.

Reviewer #2 (Remarks to the Author):

The authors have satisfactorily answered all the questions and thoroughly revised the manuscript. Therefore I recommend this paper for publication in Nature Communications.

Reviewer #3 (Remarks to the Author):

The authors have adequately responded to my remaining questions, as well as those of the other reviewer. It is my opinion that the manuscript can now be published without further revision.

Responses to Reviewers' Comments

Reviewer #1

The authors have conducted a study on a highly conducting solid electrolyte, solid solution of Li(CB₉H₁₀) and Li(CB₁₁H₁₂), and present valuable results on its application in solid-state lithium-sulfur batteries. With this electrolyte, they have obtained very high ionic conductivities at room temperature, together with a very good stability against lithium and outstanding cell performances. As opposed to already published study, on almost the same type of electrolyte, the new proposed stoichiometry ensure the high temperature phase stability at room temperature. The number of cycle test, 100, albeit at 60 C but at rather high charge / discharge rates, is very encouraging for practical application of the batteries. In the revised version of the manuscript, the authors have taken careful care of the suggestions and comments from the reviewers. In their rebuttal, they have properly and soundly address the different remarks and questions.

Response: We are grateful to the reviewer for his/her comprehensive review of our manuscript and positive evaluation.

1. The only weak point may concern their evaluation of the grain size for the ball-milled powder. The author do not explain how they actually evaluated the grain size. If, it is based on the Scherrer equation, then the value given is certainly wrong as it is commonly accepted that this equation is valid for crystallites with sizes up to 1 micrometer at max. I recommend the authors, either to prove the validity of the reported value or to delete it from the manuscript and supplementary information. Their addition concerning the conductivity measurement of the ball milled Li(CB₉H₁₀) appears to be sufficient to answer the reviewer's #3 question 1). More interestingly, from Figure 8, one can see the sharpening of the peaks after the heat treatment. It may be related to some annealing of the defect caused by the ball milling. If the authors could provide a conductivity measurement for this sample, this will certainly close the case.

Response: We found this reviewer's point extremely helpful in improving the quality of the manuscript. This is much appreciated. As the reviewer rightly pointed out, the grain size was estimated by the Scherrer equation, which was not appropriate for the evaluation of our materials. Therefore, in accordance with the reviewer's suggestion, the conductivity of heat-treated 0.7Li(CB₉H₁₀)-0.3Li(CB₁₁H₁₂) was measured, as shown below. The impedance results indicate that the heat-treated 0.7Li(CB₉H₁₀)-0.3Li(CB₁₁H₁₂) exhibited no change in terms of conductivity compared to the pristine 0.7Li(CB₉H₁₀)-0.3Li(CB₁₁H₁₂), which indicates that the effect of size is negligible. To reflect this important point, the sentence describing grain size was deleted, and the impedance results for the heat-treated 0.7Li(CB₉H₁₀)-0.3Li(CB₁₁H₁₂) were added to Fig. S10b. The relevant sentences were revised as follows:

[Fig. S10b in SI]

[Page 8]

...(Fig. 2b and Supplementary Fig. 10a). In contrast, for $0.7\text{Li}(\text{CB}_9\text{H}_{10})-0.3\text{Li}(\text{CB}_{11}\text{H}_{12})$, the Arrhenius plot shows no phase-transition-derived changes in conductivity and a linearly increase in the logarithmic values (Fig. 2b and Supplementary Fig. 10b).

[Page S11]

Supplementary Figure 10. Effects of ball-milling and heat-treatment on conductivity. a,b, Ionic conductivities of **(a)** $\text{Li}(\text{CB}_9\text{H}_{10})$ before and after ball-milling at 400 rpm for 20 h and **(b)** $0.7\text{Li}(\text{CB}_9\text{H}_{10})-0.3\text{Li}(\text{CB}_{11}\text{H}_{12})$ before and after heat-treatment at 473 K for 12 h.

The impedance results of ball-milled $\text{Li}(\text{CB}_9\text{H}_{10})$ and heat-treated $0.7\text{Li}(\text{CB}_9\text{H}_{10})-0.3\text{Li}(\text{CB}_{11}\text{H}_{12})$ indicate that the size has a negligible effect on ionic conductivity.

Reviewer #2

The authors have satisfactorily answered all the questions and thoroughly revised the manuscript. Therefore I recommend this paper for publication in Nature Communications.

Response: We thank the reviewer very much for his/her positive evaluation.

Reviewer #3

The authors have adequately responded to my remaining questions, as well as those of the other reviewer. It is my opinion that the manuscript can now be published without further revision.

Response: We thank the reviewer very much for his/her positive evaluation.